

# Associations with monetary values do not influence access to awareness for faces

Marcus Rothkirch[1], Maximilian Wieser[1] and Philipp Sterzer[1,2]

[1] Charité –Universitätsmedizin Berlin, corporate member of Freie Universität Berlin and Humboldt-Universität zu Berlin, Department of Psychiatry and Psychotherapy, Berlin, Germany
[2] Bernstein Center for Computational Neuroscience, Humboldt Universität Berlin, Berlin, Germany

## ABSTRACT

Human faces can convey socially relevant information in various ways. Since the early detection of such information is crucial in social contexts, socially meaningful information might also have privileged access to awareness. This is indeed suggested by previous research using faces with emotional expressions. However, the social relevance of emotional faces is confounded with their physical stimulus characteristics. Here, we sought to overcome this problem by manipulating the relevance of face stimuli through classical conditioning: Participants had to learn the association between different face exemplars and high or low amounts of positive and negative monetary outcomes. Before and after the conditioning procedure, the time these faces needed to enter awareness was probed using continuous flash suppression, a variant of binocular rivalry. While participants successfully learned the association between the face stimuli and the respective monetary outcomes, faces with a high monetary value did not enter visual awareness faster than faces with a low monetary value after conditioning, neither for rewarding nor for aversive outcomes. Our results tentatively suggest that behaviorally relevant faces do not have privileged access to awareness when the assessment of the faces' relevance is dependent on the processing of face identity, as this requires complex stimulus processing that is likely limited at pre-conscious stages.

## INTRODUCTION

The ability to identify and to rapidly read information from human faces has a pivotal role in social contexts. Since the multitude of information conveyed by faces goes far beyond the image per se, different cognitive systems are involved in face processing. Faces are thus a popular tool to assess what types of information can be processed without the observer's awareness or have preferential access to awareness (*Axelrod, Bar & Rees, 2015*; *Madipakkam & Rothkirch, 2019*). A particular focus of previous research in this context was on the question whether the social meaning of faces is already processed at pre-conscious stages, thereby facilitating conscious awareness of faces that convey socially relevant information. Indeed, facial cues signaling threat (*Yang, Zald & Blake, 2007*; *Yang & Yeh, 2018*), trustworthiness (*Stewart et al., 2012*; *Getov et al., 2015*), or positive emotions (*Stein & Sterzer, 2012*) seem to accelerate the awareness of faces. However, emotional

Corresponding author
Marcus Rothkirch,
marcus.rothkirch@charite.de

expressions and other social characteristics of faces, such as trustworthiness or dominance, are inextricably linked to physical stimulus properties, like contrast, luminance, or spatial frequencies. In fact, the prioritization of emotional faces can be largely explained by differences in such physical stimulus properties (*Stein & Sterzer, 2012*; *Gray et al., 2013*; *Hedger et al., 2016*; *Stein et al., 2018*). To be able to unequivocally attribute differences in the access to awareness of faces to their behavioral relevance, however, the influence of physical stimulus properties should be ruled out first (*Moors et al., 2019*).

An elegant way to circumvent the inherent confound between physical stimulus properties and higher-level relevance is to ascribe behavioral relevance to faces in a systematic and controlled manner. That way, the association between the physical characteristics and the relevance of stimuli can be balanced out across observers. *Anderson et al. (2011)* followed such an approach by pairing faces with positive, negative, or neutral gossip. They observed that faces previously paired with negative gossip dominated visual awareness during a following binocular rivalry task. In subsequent studies, in contrast, affective biographical information did not influence observers' awareness of faces, suggesting a rather limited impact of such information on visual awareness (*Rabovsky, Stein & Abdel Rahman, 2016*; *Stein et al., 2017*). One reason for these conflicting findings might be that the relevance of social information depends on each individual's evaluation of this information. Indeed, the time for complex stimuli to reach awareness can depend on the subjectively experienced value of the stimulus (*Schmack et al., 2016*) or certain personality traits of the observer (*Madipakkam et al., 2019*).

In the present study, we chose to systematically pair images of faces with high or low amounts of monetary reward and punishment. Manipulating the behavioral relevance of faces by means of monetary incentives has several advantages over using verbal descriptions. In comparison to the latter, monetary values are quantitative, which implies that different conditions can be clearly defined. In this regard, different conditions are set by different amounts of the same unit, whereas for biographical information descriptions of positive behaviors or traits, for example, are compared to entirely different descriptions that are supposed to be identified as neutral. Thus, monetary values allow for a systematic control of the behavioral relevance of faces and are intersubjectively meaningful. The association of stimuli with monetary incentives by means of classical conditioning can modulate the subjective value of the stimuli to a similar extent as to primary reinforcers (*Delgado, Labouliere & Phelps, 2006*; *Lehner et al., 2016*). Moreover, such associations have the potency to influence visual attention, such that stimuli associated with higher monetary values capture and guide attention more strongly than stimuli associated with lower values (*Austin & Duka, 2010*; *Bucker & Theeuwes, 2017*; *Bucker & Theeuwes, 2018*). It is reasonable to assume that such learned associations can also affect how quickly stimuli enter awareness, since stimuli previously paired with high monetary reward are more often consciously perceived during rapid sequences of visual stimuli than stimuli paired with low reward (*Leganes-Fonteneau, Scott & Duka, 2018*). In our study, participants had to learn the association between face stimuli and monetary values. Before and after the learning phase, the same faces were presented under breaking continuous flash suppression (bCFS) to measure the time they require to get access to awareness. BCFS is a variant of binocular

rivalry, in which two different stimuli are presented to the two different eyes. Unlike binocular rivalry, however, for bCFS a dynamic stimulus (i.e., the masking stimulus) is presented to one eye, while a static stimulus (i.e., the target) is presented to the other eye. This results in the initial dominance of the masking stimulus, allowing for a greater control of stimulus suppression in comparison to the rather stochastic nature of perceptual states during binocular rivalry. We hypothesized that if learned values facilitate faces' access to awareness, there should be a stronger decrease in response times after the conditioning session for faces associated with a high monetary compared to a low monetary value. The inclusion of monetary reward as well as punishment further enabled us to detect valence-specific effects of learned values on visual awareness.

## MATERIALS & METHODS

### Participants

Twenty-four participants (13 females; age: 18–35 years, $M = 24.54 \pm 4.36$ standard deviation(SD)) took part in the experiment. Five participants were left-handed, all other participants were right-handed. All participants had normal or corrected-to-normal vision and written informed consent was obtained from each participant prior to their participation in the experiment. The study was approved by the local ethics committee of the Charité –Universitätsmedizin Berlin (EA1/301/13) and performed in accordance with the Declaration of Helsinki. The final sample size was based on a sequential testing approach using Bayes Factors (*Schönbrodt et al., 2017*). More specifically, we planned to calculate Bayes Factors (BF10) for our main analysis after every new batch of 24 participants, since this sample size is needed to completely counterbalance the association between stimulus exemplars and monetary values across participants. Our aim was to continue data collection until these Bayes Factors would either exceed a threshold of 3, which indicates evidence in favour of the alternative hypothesis, or fall below a threshold of 1/3, indicating evidence in favour of the null hypothesis. Since this condition was already met after the inclusion of 24 participants, we stopped data collection at this stage.

### Stimuli and apparatus

The face stimuli used in the study were grey-scale photographs of four different female faces with a neutral expression, taken from the NimStim Set of Facial Expressions (*Tottenham et al., 2009*; image IDs: 01, 07, 09, 17). All four images were similar in global contrast (root mean square contrast between 0.16 and 0.20) and luminance (mean luminance between 27.49 cd/m$^2$ and 31.35 cd/m$^2$). All stimuli were presented on a uniformly grey background (30.28 cd/m$^2$).

Participants viewed the screen through a mirror stereoscope providing separate visual input to the two eyes. Each participant's head was stabilized by a chin rest at a viewing distance of 60 cm. All stimuli were presented using MATLAB (The MathWorks, Natick, MA, USA) and Psychtoolbox-3 (http://psychtoolbox.org/) on a 19 inch CRT monitor (resolution: 1024 × 768 Px, refresh rate: 60 Hz).

## Procedure

The experiment consisted of three phases: (1) a pre-conditioning phase to measure baseline response times, (2) a conditioning phase during which different faces were paired with monetary outcomes, and (3) a post-conditioning phase that was identical to the pre-conditioning phase, intended to assess the change in response times after the conditioning had taken place.

In the initial pre-conditioning phase (Fig. 1A), different face exemplars were presented under continuous flash suppression. At the beginning of each trial, a black rectangle (10° × 10°) and a black fixation cross in its center (0.68° × 0.68°) were presented to each eye. The rectangle and the cross were visible throughout the whole experimental phase. After a fixation duration of 2 s, high-contrast dynamic mask stimuli consisting of circles and squares of various colors and sizes were flashed to a randomly selected eye at a frequency of 10 Hz. Simultaneously, a face image (3.75° × 3.75°) that was located within one of the four quadrants of the black rectangle was presented to the other eye. The contrast of the face stimulus linearly increased from 0% to 100% during the initial 2 s. After that, the face remained at full contrast until the end of the trial. A trial ended when the participant gave a manual response or, if no response was made, after 15 s. Participants' task was to indicate the location of the face, that is, the quadrant in which the face appeared, as fast and as accurately as possible by pressing one of four designated keys on the keyboard. This part of the experiment comprised 96 trials. The combination of the face exemplar, the location of the face, and the eye to which the face was presented was counterbalanced and randomized across trials.

In the second phase, participants had to learn the association between the face exemplars and monetary outcomes by means of classical conditioning (Fig. 1B). In each trial, one of the faces that were already presented in the first part of the experiment was shown in the center of the screen. Below the face, a positive or negative monetary value was displayed. The face and the value were presented for 5 s. After the offset of the face and the associated monetary value, a blank screen was presented for a randomized interval between 1 s and 1.6 s before the next trial started. Each face was associated with one of four different monetary values: −2 €, −0.1 €, +0.1 €, and +2 €. In 75% of the trials, the face was depicted with its associated monetary value. In the remaining trials, an outcome of 0 € was presented along with the face. Such a probabilistic reinforcement schedule was chosen to maintain participants' attention to the stimuli and the task, as for classical conditioning with monetary outcomes attention is preferably directed towards partially predictive stimuli in comparison to fully predictive stimuli (*Austin & Duka, 2010*). Participants were instructed to passively view the stimuli and to memorize the association between the faces and the monetary values as well as possible. They were further informed that the monetary value depicted in a trial would be counted towards their overall payoff. Thus, for a trial, in which a face was presented together with '+2 €', for instance, 2 € would be added to their payoff, while for trials with '-2 €' 2 € would be subtracted from their payoff. Since unbeknown to participants each monetary value was presented equally often, the outcome for this part of the conditioning phase amounted to zero. Thus, participants' outcome was solely defined by their accuracy in the query trials (see below). This phase of the experiment

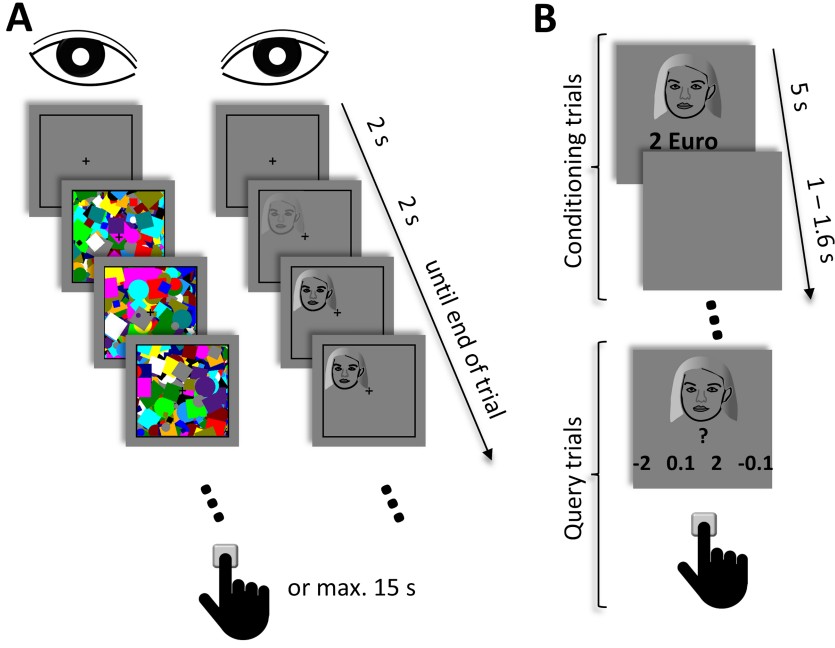

**Figure 1** **Schematic depiction of the experimental procedure.** (A) In the pre- and post-conditioning phase, high-contrast dynamic mask stimuli were presented to one eye at a frequency of 10 Hz. A face stimulus was simultaneously presented to the other eye. The contrast of the face stimulus linearly increased during the initial 2 s and remained at full contrast until the end of the trial. Participants' task was to indicate the location of the face. A trial ended either after a manual response or at the latest after 15 s. (B) In the conditioning phase, different face stimuli were presented together with their associated monetary outcome. Participants' task was to passively view and memorize these associations. This phase comprised four blocks. At the end of each block, query trials where performed, where each face stimulus was presented with four different response options. Here, participants had to select the value that was previously associated with the face. Due to copyright reasons, the original face images used in the experiment are not shown in the figure but replaced by a schematic drawing.

comprised four blocks, during which each face was presented four times. The order of the face exemplars was randomized and the association between the face exemplars and the monetary values was fully counterbalanced across participants. The association was further kept constant across all four blocks.

To assess whether participants were indeed able to learn these associations, query trials were added at the end of each block. Each face was presented once during these query trials. In contrast to the conditioning trials, a question mark and all four different monetary values were displayed below the face in random order. Participants were required to select the value that was associated with the respective face by pressing one of four designated buttons. The face and the monetary values were presented until a response was made. Participants were informed that monetary reimbursement for their participation in the experiment would depend on the accuracy of their choices during these query trials. After the experiment, participants indeed received the amount of money that they accumulated during the conditioning phase. More specifically, a correct assignment of a monetary reward to the respective face would yield an addition of +2 €or +0.1 €, respectively, to

their payoff. A wrong response to faces associated with reward did not result in a monetary gain. For faces associated with a monetary punishment, participants had to assign the correct monetary value to avoid a monetary loss. This means that a correct assignment of −2 € or −0.1 € to the respective face did not yield a monetary gain or loss. However, if a participant assigned a wrong value to a punishment-related face, this resulted in a loss of −2 € or −0.1 €, respectively. Thus, participants expected that their payoff was dependent on both, the passively viewed pairings of monetary values and faces as well as their performance during the query trials. This procedure served two purposes: (1) to increase the relevance of the high-valued compared to the low-valued faces, and (2) to increase participants' attention to all face stimuli during the query trials so that the accuracies in the query trials would provide an optimal account of participants' learning progress. Note that in case of negative values participants had to respond accurately to avoid monetary losses during the query trials.

The post-conditioning phase, which followed directly after the conditioning phase, was fully identical to the pre-conditioning phase. After the experiment, participants rated how much they felt motivated by the different monetary values to memorize the faces on a visual analog scale ranging from 0 to 5.

## Data analysis

Participants' learning performance during the conditioning phase was assessed on the basis of their responses in the query trials. For each of the four blocks, each participant's accuracy in assigning the monetary value to each face exemplar was computed. The chance level for each block was .25. To evaluate participants' sensitivity for reward and punishment, we computed each participant's response bias during the query trials. For reward-related faces the response bias is computed on the basis of the z-transformed hits ($H_R$) and false alarm rates ($FA_R$) as follows:

$$c_R = -\frac{z(H_R) + z(FA_R)}{2}$$

However, since participants have to assign a positive or negative value to each face, the response biases for reward and punishment are not independent of each other. More specifically, an increase in the false alarm rate for reward leads to a decrease in the hit rate for punishment ($H_P$) by the same amount, such that: $z(FA_R) = -z(H_P)$. Thus, the formula above can be rewritten as:

$$c_R = -\frac{z(H_R) - z(H_P)}{2}.$$

Since a bias towards reward ($c_R$) automatically implies a bias away from punishment ($c_P$) such that $c_R = -c_P$, we only report the reward response bias. For $c_R$, a positive bias denotes a stronger sensitivity for reward, while a negative bias denotes a stronger sensitivity for punishment.

For the bCFS phase before and after the conditioning phase, trials in which participants responded incorrectly (percentage of trials: $M = 3.19\%$, $SD = 2.75\%$) or failed to give a response until the end of a trial ($M = 0.61\%$, $SD = 1.26\%$) were discarded from further analysis. Furthermore, response times below 200 ms were considered anticipatory

responses and were also discarded from further analysis (percentage of trials: $M = 0.61\%$, $SD = 1.26\%$). To compare the response times between the different conditions, we followed two approaches. For the first approach, we computed the median response time for each participant and condition before and after conditioning. We used the median instead of the mean to account for the skewness of response time distributions, which is line with other bCFS studies (e.g., *Gayet et al., 2016*; *Gayet & Stein, 2017*; *Han, Blake & Alais, 2018*). We then subtracted the median response times of the pre-conditioning phase from the median response times of the post-conditioning phase, which resulted in a measure for the change of response times for each condition and participant. Finally, we performed two paired $t$-tests to compare the mean change in response times between high and low reward as well as between high and low punishment. The alpha level of these two t-tests was adjusted to .025 to account for multiple comparisons. For the second approach, we analyzed the response time distributions in more depth by computing hierarchical shift functions (*Rousselet & Wilcox, 2019*), which can be more sensitive to response time differences, especially when they are restricted to early or late responses. To this end, we computed the deciles of the response time distribution for each condition and participant before and after conditioning. In a next step, we subtracted the deciles of the pre-conditioning phase from the deciles of the post-conditioning phase. The resulting values thus indicate the change in response times for each segment of the whole distribution, where lower deciles reflect faster responses and higher deciles slower responses. For each decile, we then performed a paired $t$-test to compare the changes in response times of the high reward to the low reward condition and of the high punishment to the low punishment condition. As this amounts to 18 different t-tests in total, we adjusted the alpha level of the $t$-tests to .003. Thus, we adjusted the significance level for the number of tests across reward and punishment, because our aim was to control the maximum experiment-wise error rate (*Bender & Lange, 2001*). Our reasoning for applying such a rigorous adjustment of the alpha level was that the underlying hypothesis of an influence of learned values on bCFS response times would be supported by a difference in response times for either reward or punishment.

To probe potential time-dependent effects of learned values, we additionally split the post-conditioning phase into two halves. For statistical inference, we performed repeated-measures ANOVAs with the factors time (first half vs. second half of the post-conditioning phase) and value (high value vs. low value). Separate ANOVAs were performed for reward-related and punishment-related stimuli. The focus of this analysis was on the interaction between time and value, since a statistically significant interaction would signify an influence of the monetary value that was dependent on time. Note that this analysis was performed on the median response time differences between the pre- and post-conditioning phase.

Finally, we performed two generalized linear mixed-effects models on the single trial data, one for reward-related and one for punishment-related stimuli. This analysis was intended to take into account the variability of the effect of interest in dependence on the repetition of the stimuli and the association between the different face exemplars and the monetary values. We defined a Gamma distribution as the distribution of our response variable, since it provides a plausible approximation to the processes reflected in response

times (*Lo & Andrews, 2015*). We included the interaction of face value, bCFS phase, face exemplar, and trial number as well as all other interactions and main effects of these variables as fixed effects in our models, for which we used effects coding. For the random effects term, we defined the theoretically maximal model by including subject as a random factor together with the by-subject random intercept and the by-subject random slopes of all fixed factors (*Barr et al., 2013*). Both models thus had the following structure:

$$RT \sim value * phase * exemplar * trial + (1 + value * phase * exemplar * trial | subject).$$

The estimate of each fixed-effects predictor was then tested against zero by means of individual $t$-tests.

In addition to frequentist inference statistics, we computed Bayes Factors (BF10) for each $t$-test, using a default Cauchy prior of 0.707. For ANOVAs we computed BF10 directly from the $F$-values of the ANOVA statistics (*Faulkenberry, 2018*). In line with previous suggestions (*Wetzels & Wagenmakers, 2012*), we interpret BF10 >3 as evidence for the alternative hypothesis and BF10 <1/3 as evidence for the null hypothesis. We also report BF01 to clarify which analyses yielded evidence for the null hypothesis. In this case, BF01 >3 indicates evidence for the null hypothesis.

## RESULTS

### Classical conditioning

At the end of every block in the conditioning phase, participants had to indicate the associated monetary value for each presented face. Figure 2A depicts the accuracy of these responses for the four different blocks. Since participants had to make four choices in each block, the chance level corresponds to an accuracy of .25 for each block. Binomial tests indicated that the response accuracies across all four blocks exceeded chance level for all participants ($p \leq .002$, BF10 $\geq$ 10.10). Figure 2B shows the response accuracies across all four blocks separately for each value condition. As for each value condition the majority of participants did not commit any mistake during the query trials, the median accuracy for each condition amounted to 1. Thus, participants quickly and successfully learned the associations between the face exemplars and the monetary outcomes for each of the different monetary values. After the experiment, participants rated their motivation for each monetary value on a visual analog scale ranging from 0 to 5. In comparison to a low reward of 0.1 € ($M = 1.51 \pm 0.28$ standard error of the mean [SEM]), participants felt substantially more motivated by the high reward of 2 € ($M = 2.98 \pm 0.36$ SEM; paired $t$-test: t(23) = 4.99, $p <$ .001, BF10 = 521.53). The difference in motivation between a low punishment of 0.1 € ($M = 1.46 \pm 0.30$ SEM) and a high punishment of 2 € ($M = 2.08 \pm 0.37$ SEM) was less pronounced compared to reward (t(23) = 2.42, $p = .024$, BF10 = 2.35), but still statistically significant at a corrected threshold of $\alpha = .025$. The majority of participants ($n = 18$) did not exhibit a response bias to either reward or punishment (i.e., $c_R = 0$). One participant showed a response bias towards monetary punishment ($c_R = -0.18$) and five participants showed a response bias towards monetary reward ($c_R = 0.24$–0.97).
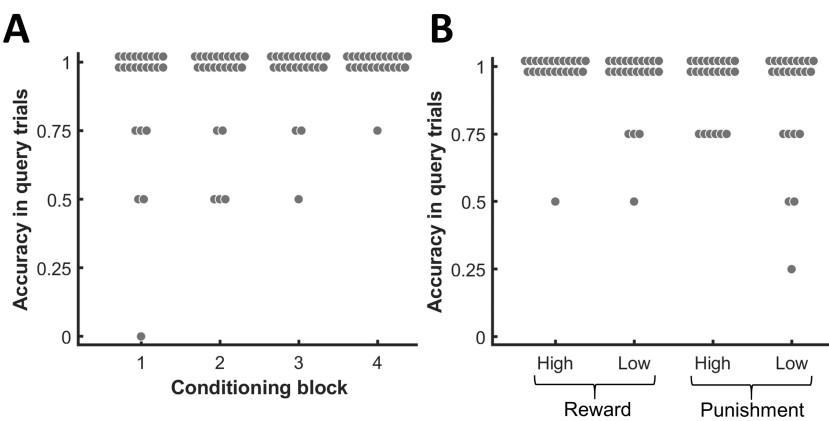

**Figure 2** **Choice accuracy in the query trials of the conditioning phase.** (A) Accuracy for each block of the conditioning phase. (B) Accuracy for each value condition. Each dot represents one participant.

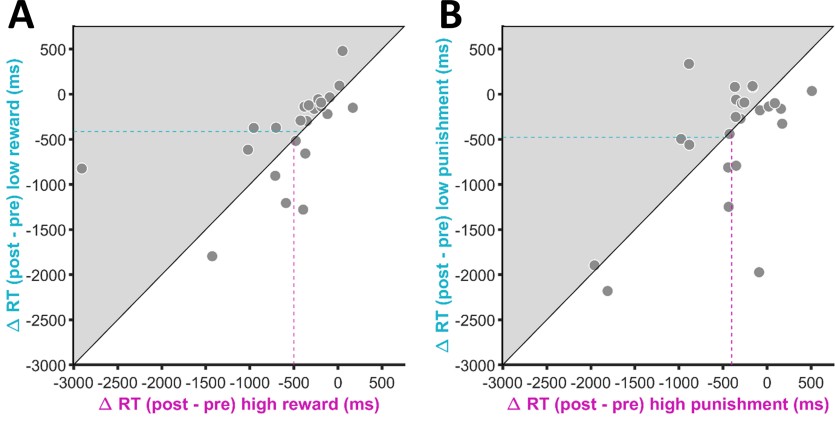

**Figure 3** **Change in response times (RT) for (A) the reward-related and (B) the punishment-related stimuli.** Negative values indicate an RT decrease in the post-conditioning phase compared to the pre-conditioning phase. The position of each data point is defined by the RT change for the high value condition on the $x$-axis and the RT change for the low value condition on the $y$-axis. Participants who showed a stronger RT reduction for the high in comparison to the low value condition (i.e., RT differences consistent with our a priori hypotheses) are located in the grey-shaded area. The magenta-coloured dashed line indicates the mean RT change for the high value condition across participants. The turquoise dashed line indicates the mean RT change for the low value condition across participants.

## Response times during breaking-CFS

Figure 3A shows the change in response times from the pre-conditioning to the post-conditioning phase for the two different reward conditions. For the high reward condition, response times decreased by 501.01 ms ($\pm$102.27 ms SEM), on average, while for the low reward condition we observed an average decrease in response times of 412.74 ms ($\pm$84.25 ms SEM). The difference between the two conditions was not statistically significant (t(23) = 0.80, $p$ = .43) and the Bayes factor indicated the absence of an effect (BF10 = 0.29, BF01 = 3.48).

The average change in response times for the high and low punishment condition are depicted in Fig. 3B. Numerically, there was a stronger decrease in response times for the low punishment condition ($M = -477.85 \pm 97.54$ ms SEM) in comparison to the high punishment condition ($M = -401.97 \pm 82.05$ ms SEM). However, the difference between the two conditions was again not statistically significant and indicative of an absence of an effect (t(23) = 0.65, $p = .52$, BF10 = 0.26, BF01 = 3.84).

To rule out baseline differences between conditions that might have masked potential effects of learned values in the post-conditioning phase, we additionally computed response times during the pre-conditioning phase only. Response times for faces that were later paired with a high reward ($M = 2275.90 \pm 464.57$ ms SEM) did not significantly differ from response times for faces that were later paired with a low reward ($M = 2254.57 \pm 460.21$ ms SEM; t(23) = 0.17, $p = .87$, BF10 = 0.22, BF01 = 4.60). Similarly, the time to respond to faces later associated with high punishment ($M = 2224.52 \pm 454.08$ ms SEM) was not significantly different from faces later associated with low punishment ($M = 2315.77 \pm 472.71$ ms SEM; t(23) = 0.85, $p = .40$, BF10 = 0.30, BF01 = 3.36).

Differences between high and low values, however, might be limited to a specific range of the whole response time distributions, that is, the learned values could specifically affect fast or slow responses, which would not necessarily be captured by measures of central tendencies. We therefore explored the response time distributions in more depth by computing the change between the pre- and the post-conditioning phase for each decile of the whole response time distribution for each condition. For reward-related stimuli (Fig. 4A) as well as well as for punishment-related stimuli (Fig. 4B), the decrease in response times was more pronounced for slower responses. However, there was no significant difference between high and low values for any of the deciles ($p \geq .12$ and BF10 $\leq 0.65$ for reward, $p \geq .12$ and BF10 $\leq 1.66$ for punishment). Of note, the differences between high and low reward and punishment, respectively, were close to an uncorrected significance level of .05 for the last deciles. The response time decreases in this decile, however, were numerically stronger for low values compared to high values and as such contrary to our a priori hypothesis.

Due to extinction, potential influences of the monetary associations on response times could have quickly decayed during the post-conditioning phase. Thus, averaging across all responses in the post-conditioning phase might have masked the effects of learned values. To identify potential time-dependent effects, we split the post-conditioning phase into two halves. We performed two repeated-measures ANOVAs with the factors time (first half vs. second half of the post-conditioning phase) and value (low value vs. high value). If the influence of learned values was indeed dependent on time such that it quickly faded after the conditioning block, this should be reflected in the interaction between time and value. However, neither for faces previously associated with reward (F(1,23) = 0.60, $p = .44$, BF10 = 0.28, BF01 = 3.59) nor for faces associated with punishment (F(1,23) = 0.76, $p = .39$, BF10 = 0.30, BF01 = 3.31) we observed an interaction between time and value.

In line with our main analysis, the parameter estimates for the interaction of the face value and the bCFS phase in the generalized linear mixed-effects models were not statistically significant from zero, neither for reward ($\beta = -8.49 \times 10^{-6}$, t(2194) = -1.87$\times 10^{-5}$, $p > .99$)

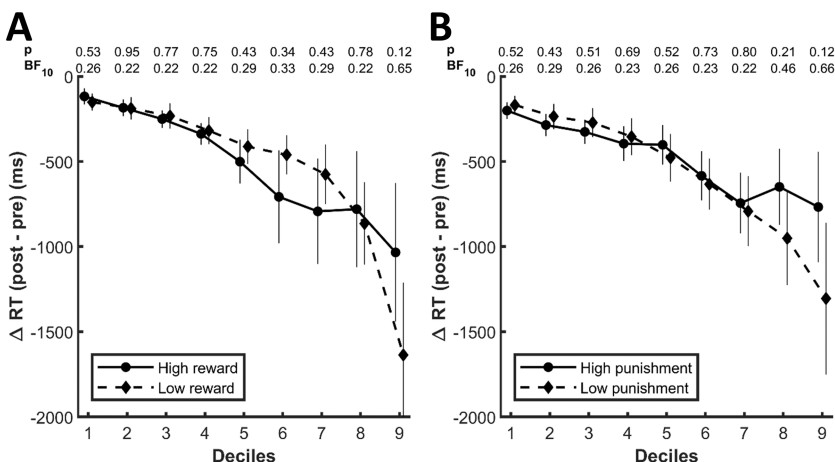

**Figure 4** **Change in response times for each decile of the whole response distributions for the (A) reward and (B) punishment conditions.** The numbers at the top of each graph indicate the *p*-values and the Bayes Factors for the comparison between the two conditions for each respective decile. The error bars display standard errors of the mean.

nor for punishment ($\beta = -4.26{*}10^{-5}$, t(2193) $= -8.62{*}10^{-5}$, $p > .99$). There were also no other statistically significant interactions with trial number or face exemplar (all $p > .99$).

## Exploratory analyses

While our a priori hypotheses focused on the comparison of the change in response times for high vs. low monetary values, we performed two additional exploratory analyses to examine whether the learned values might have influenced the response times during the bCFS phase in other ways.

For the first analysis, we collapsed our data across high and low values and compared response time changes for all reward-related faces to all punishment-related faces. For reward-related faces, response times decreased by 428.13 ms ($\pm$87.39 SEM), while for punishment-related faces response times decreased by 443.98 ms ($\pm$90.63 SEM), on average. The difference between reward and punishment was not statistically significant (t(23) $= 0.19$, $p = .85$, BF10 $= 0.22$, BF01 $= 4.59$). Even when only faces associated with a high value (i.e., high reward vs. high punishment) were considered, the difference between reward and punishment did not reach statistical significance (t(23) $= 1.15$, $p = .26$, BF10 $= 0.39$, BF01 $= 2.58$).

In a second exploratory analysis we examined whether the influence of learned values on response times might have been dependent on the learning performance during the conditioning phase. To this end, we computed each participant's accuracy in the query trials across all four conditioning blocks. We then correlated these accuracies against the response time differences (i.e., post- vs. pre-conditioning) during the bCFS blocks, separately for reward- and punishment-related faces. The bCFS response time differences were computed by subtracting the change in response times for the low value condition

from the change in response times for the high value condition as follows:

$$([RT_{post} - RT_{pre}]_{high\ reward} - [RT_{post} - RT_{pre}]_{low\ reward})\ and$$

$$([RT_{post} - RT_{pre}]_{high\ punishment} - [RT_{post} - RT_{pre}]_{low\ punishment}).$$

As such, negative values indicate a change in response times in line with our a priori hypotheses, that is, a stronger decrease in response times for the high compared to the low value condition. Since only few participants exhibited low accuracies during the query trials, the distribution of participants' accuracies was heavily skewed. To account for this, we computed Spearman's $\rho$. Neither for reward ($\rho = .30$, $p = .16$, BF10 = 0.43, BF01 = 2.35) nor for punishment ($\rho = .13$, $p = .56$, BF10 = 0.19, BF01 = 5.39), accuracies were related to response time differences between conditions. Note that if higher accuracies in the query trials were related to a greater influence of high monetary values on response times, this would be indicated by a negative correlation. Due to the high accuracy during the query trials in the majority of participants, this analysis should be interpreted with caution.

## DISCUSSION

We studied whether learned values, established by means of classical conditioning with monetary reward and punishment, influence access to awareness for faces. While participants successfully learned the association between the different face exemplars and the monetary values, the learned association did not have an influence on their response times. Response times generally decreased from the pre- to the post-conditioning phase. However, this decrease was equally strong for high compared to low reward and for high compared to low punishment. A more in-depth exploration of the response time distributions did not reveal an advantage for faces paired with a higher compared to a lower monetary value either.

Faces can express and convey their relevance in various ways, for instance through their emotional expression or particular facial features such as eye gaze. In the current study, we paired faces with monetary incentives to render them behaviorally relevant. This way, we intended to circumvent a confound between physical stimulus characteristics and higher-level social relevance, which is especially prevalent in the investigation of the awareness of emotional faces (*Hedger et al., 2016*). While it has previously been suggested that learned associations between affective information and faces affect the faces' potency to dominate awareness (*Anderson et al., 2011*), subsequent studies did not observe a privileged access to awareness for faces paired with affective information (*Rabovsky, Stein & Abdel Rahman, 2016*; *Stein et al., 2017*). Our findings are in line with the latter studies, while at the same time complementing these previous results by showing that not only biographical information but also monetary incentives fail to facilitate awareness of faces. The approach of associating face stimuli with affective information, however, differs in two aspects from the investigation of the influence of emotional expressions. First, emotional expressions are inherent to a face, while the association with affective information has to be learned. Secondly, the identification of emotional expressions often only requires the processing

of certain facial features. A rapid detection of fearful faces, for instance, is likely due to the greater exposure to the iris and the sclera in fearful faces (*Whalen et al., 2004*). Effects of learned associations, in contrast, likely requires the identification of the faces' whole identity. Since conditioned responses to fear-conditioned faces transfer to novel images of the same face identity (*Rehbein et al., 2018*), learned associations are indeed, at least partly, related to the face's identity. Thus, the influence of learned affective information, as in our case by means of monetary values, is dependent on a more complex analysis of the stimulus at pre-conscious stages. As the processing of face identity is rather limited under visual masking (*Moradi, Koch & Shimojo, 2005*; *Amihai, Deouell & Bentin, 2011*), the scope of pre-conscious face processing might not be sufficient to boost faces into awareness that have been coupled with positive or negative outcomes. Similar limitations likely apply to other types of complex stimuli when different exemplars of the same category are paired with different outcomes. Though to what extent pre-conscious processing is limited for different stimulus categories has to be further investigated in future studies.

Participants were clearly able to learn the association between the face exemplars and the respective monetary values. The absence of an influence of the associated values on the access of the faces to awareness can thus not be attributed to a failure or difficulties of participants to learn these associations. There is ample evidence for sustained neural (*Rothkirch et al., 2012*) and behavioral effects (*Raymond & O'Brien, 2009*; *Rutherford, O'Brien & Raymond, 2010*; *Rothkirch et al., 2013*) of previously learned associations between faces and monetary values, indicating that pairing face stimuli with monetary outcomes has the potency to render face stimuli behaviorally relevant for an extended period of time. It must be noted, however, that such associations have been mostly induced by means of instrumental conditioning so far. While classical conditioning with monetary incentives can generally bring about similar effects compared to instrumental conditioning (*Delgado, Labouliere & Phelps, 2006*; *Bucker & Theeuwes, 2016*; *Bucker & Theeuwes, 2017*), the specific combination of face stimuli and monetary outcomes in the context of classical conditioning has only rarely been studied so far. However, *Trilla Gros, Panasiti & Chakrabarti (2015)* found that EEG responses to faces are influenced by their previous associations with monetary reward and punishment established through classical conditioning. While this shows that classical conditioning with monetary values has the potency to alter neural signals in response to face stimuli, it leaves open the question how such altered neural signals translate into behavioral effects. It has further been demonstrated previously that simple visual stimuli, like gratings, can gain faster access to awareness by means of classical fear conditioning (*Gayet et al., 2016*), showing that a conditioning approach can, in principle, confer behavioral relevance to visual stimuli such that they access awareness more rapidly. There is the residual possibility that the association between the face stimuli and the monetary values in our study did not effectively change the affective content of the faces, especially because associating faces with monetary incentives by means of classical conditioning might be less effective compared to instrumental conditioning. For fear-conditioning, the effectiveness of the conditioning procedure is usually assessed on the basis of physiological measures, like skin conductance responses. For appetitive conditioning and conditioning with monetary outcomes, in contrast, such a standard

physiological measure has not yet been established. Since pupil size promises to be a fruitful measure of the effectiveness of appetitive conditioning (*Pietrock et al., 2019*), it could be assessed in future studies focusing on the influence of learned stimulus values on the access to awareness. The availability of such a measure would also allow to relate the strength of conditioning in each individual to the effect of learned values on visual awareness (*Madipakkam et al., 2016*; *Vieira et al., 2017*). Alternatively or complementary to physiological responses, the effectiveness of the conditioning procedure could further be evaluated on the basis of a control task with clearly visible stimuli. Such a control task, though, would either require a separate sample of participants or would have to be included after the conditioning phase in addition to the bCFS task, in which case the extinction of the conditioned values plays an important role. Physiological measures, in contrast, could be more easily integrated in the existing experimental design.

A further relevant aspect are potential asymmetries in participants' sensitivity to reward or punishment. We assessed participants' sensitivity in our study on the basis of their responses during the query trials in the conditioning phase. Overall, we observed a slightly greater sensitivity towards reward compared to punishment, suggesting that for some participants seeking rewards was more relevant during the conditioning phase than avoiding punishment. However, the overwhelming majority of participants did not exhibit a bias in any direction and almost all participants were able to correctly assign the monetary values to the different face exemplars until the end of the conditioning phase. Notably, in case of negative values participants had to respond accurately to avoid monetary losses during the query trials. It is possible that this task has rendered the loss-related faces similar to the reward-related faces in terms of their motivational content. It has indeed been argued previously that the avoidance of aversive outcomes can be rewarding (*Kim, Shimojo & O'Doherty, 2006*), which would imply that the loss-related faces in our study might have been perceived as motivationally positive. This assumption is in conflict with several other studies, however, indicating that separate neural structures underlie reward and avoidance learning (*Yacubian et al., 2006*; *Palminteri et al., 2012*; *Rothkirch et al., 2017*), which suggests that the avoidance of punishment is qualitatively distinct from receiving a reward.

For the assessment of possible asymmetries in the sensitivity between reward and punishment or motivational similarities between seeking rewards and avoiding losses, it is important to take into account that we compared response times in the bCFS phase between high and low values separately for reward and punishment. Thus, any influences that might have affected the relation between reward- and punishment-related faces could not have biased our results.

It is conceivable that an influence of learned values on response times might have been dampened either due to habituation because of the repeated exposure to each face exemplar or due to extinction during the post-conditioning phase. Furthermore, it has previously been reported that responses to fear-conditioned faces rapidly decrease when these faces are visually masked (*Raio et al., 2012*). Thus, effects of learned values can be obscured when responses are aggregated across the post-conditioning phase. However, an analysis that distinguished between the first and second half of the post-conditioning phase did not

provide any indication of such time-dependent effects in our study. Moreover, while *Raio et al. (2012)* conducted the conditioning procedure with faces that were suppressed from awareness, which likely established only unstable associations between the conditioned and unconditioned stimuli, the conditioning procedure in our study was performed with fully visible face stimuli. Still, each face exemplar was repeated 64 times across all phases in our study. Indeed, in comparison to previous studies (*Anderson et al., 2011*; *Rabovsky, Stein & Abdel Rahman, 2016*; *Stein et al., 2017*) our conditioning phase comprised more repetitions of each face exemplar and a lower number of different face exemplars. It has to be noted, however, that this ensured that participants successfully learned the associations between faces and values. *Rabovsky, Stein & Abdel Rahman (2016)*, in contrast, report that in their study only 36% of newly learned associations could be explicitly recalled after the learning phase, on average. In the other two studies (*Anderson et al., 2011*; *Stein et al., 2017*), the learning procedure was repeated until participants reached a criterion of at least 60% correct responses. Consequently, participants differed in the frequency with which they were exposed to the face stimuli. Furthermore, they underwent the post-conditioning task even though some of them might not have learned the correct associations for a substantial amount of faces until the end of the learning phase. This suggests that fewer repetitions during the learning phase likely come at the expense of a poorer learning performance, which could have also contributed to the discrepant findings in previous studies. According to *Vansteenwegen et al. (2006)*, the affective content that faces gain through classical conditioning is further largely resistant to extinction, at least at the behavioral level. In their first experiment, response times in an affective priming task were still influenced by the learned values after extinction, even though the experiment comprised 60 repetitions of each of the two different conditioned face stimuli in total, which is comparable to the 64 repetitions of each face exemplar in our study.

While we used faces with a neutral expression in our study, a potential approach to further strengthen the association between the faces and the monetary values is to use faces with an emotional expression. As suggested by the 'preparedness hypothesis', faces with different emotional expressions might be differentially prepared to become associated with different outcomes (*Dimberg & Öhman, 1996*). In this context, pairing aversive outcomes with angry faces, for instance, might be more effective than pairing them with neutral faces. The specific interactions between different emotional expressions and monetary outcomes have not been systematically studied yet, however, and such an approach my come at the expense of potential ceiling effects (*Lonsdorf et al., 2017*). Finally, while we have used monetary outcomes to render face stimuli behaviorally relevant, the use of other reinforcers, like liquid rewards in water-deprived participants or bursts of white noise, are conceivable alternatives. We chose monetary outcomes as they are easy to administer and can be equally employed as rewarding and aversive stimuli. Furthermore, the processing of primary and secondary reinforcers, including monetary values, shows large overlaps in the human brain (*Izuma, Saito & Sadato, 2008*; *Delgado, Jou & Phelps, 2011*; *Sescousse et al., 2013*), which suggests that monetary values can evoke similar positive or negative experiences in comparison to other types of reinforcers.

## CONCLUSIONS

To conclude, we did not observe a privileged access to awareness for faces that were associated with positive or negative monetary outcomes, although participants quickly learned these associations. This tentatively suggests that learned values that are tied to a face's identity have only limited influence on the face's access to awareness, as such an influence possibly exceeds the scope of pre-conscious processing.

### Funding

This work was supported by the German Research Foundation (No. RO 4836/2-1 and STE 1430/9-1). Support was also received from the German Research Foundation (DFG) and the Open Access Publication Fund of Charité - Universitätsmedizin Berlin. The funders had no role in study design, data collection and analysis, decision to publish, or preparation of the manuscript.

### Grant Disclosures

The following grant information was disclosed by the authors:
German Research Foundation: RO 4836/2-1, STE 1430/9-1.
German Research Foundation (DFG).
The Open Access Publication Fund of Charité - Universitätsmedizin Berlin.

### Competing Interests

The authors declare there are no competing interests.

### Author Contributions

- Marcus Rothkirch conceived and designed the experiments, performed the experiments, analyzed the data, prepared figures and/or tables, authored or reviewed drafts of the paper, and approved the final draft.
- Maximilian Wieser performed the experiments, authored or reviewed drafts of the paper, and approved the final draft.
- Philipp Sterzer conceived and designed the experiments, authored or reviewed drafts of the paper, and approved the final draft.

### Human Ethics

The following information was supplied relating to ethical approvals (i.e., approving body and any reference numbers):

The study was approved by the local ethics committee of the Charité – Universitätsmedizin Berlin (Ethical Application Reference: EA1/301/13).

### Data Availability

Raw data and code are available in the Supplemental Files.

## Supplemental Information

Supplemental information for this article can be found online at http://dx.doi.org/10.7717/peerj.10875#supplemental-information.

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
