# Peer review of "Associations with monetary values do not influence access to awareness for faces"

_PeerJ, doi:10.7717/peerj.10875_

## Round 0.1 · original submission · Major Revisions

The Reviewers concur in finding the study interesting and the manuscript well-written. However, they have numerous major concerns that need to be addressed.

The first issue that has been raised by Reviewers #2 and #3 and I strongly agree with them, is that the sample size is very small and, most importantly, not justified by any prospective power analysis. Although using a Bayesian hypothesis testing permits to compare the relative support for a hypothesis conditioning on the data in hand, concerns about the design efficiency remain, and the authors should discuss them more extensively. Related to reporting the BFs, I think that, in the presence of more relative evidence for the null, it is clearer to report BF01 instead of BF10.

The issues about sample size are intensified by the great flexibility in the analytic approach, as not always the analysis seems the be suited to testing the hypotheses, as pointed out by all the Reviewers. I also concur with Reviewer #3, who finds the removal of the outlier troublesome without any objective cut-off for defining such an observation as such. Furthermore, the authors should bear in mind that multivariate outliers (influential observations) rather than univariate outliers pose a threat to the validity of the inference.

Although the study design elegantly tests the authors' hypotheses, both Reviewer #1 and Reviewer #3 question the construct validity of the authors' manipulation (i.e., the efficacy of the conditioning procedure); this is a very important point to address.

Reviewers #2 and #3 also suggest taking accuracy into account. I think that perhaps a multilevel generalized linear model (link function = logit) on accuracy may provide some more information. For instance, it might be that when participants do not respond in time, that might be due to a relevant experimental effect on awareness.

Reviewers #1 and #3 both report their concerns about using only four identities. Furthermore, Reviewer #1 suggests reporting the trial number in order to allow for more fine-grained analyses on time-dependent effects. Perhaps a possible analytical approach to consider any variability in the experimental effect conditional on the face identity and the effect of repetition the authors may try to perform a multilevel linear model on the long format data, with the trial number as a continuous and face identity as a categorical predictor. Then the authors can model participants as grouping factors for modeling varying (random) intercepts and varying (random) effects. However, face identity should be better modeled as a constant (fixed) effect, as a random effect should have at least five levels to be reliably estimated.

As a somewhat minor point, when reporting dispersion indices for descriptive purposes (e.g., page 4, lines 97-98), it is better to report standard deviations rather than SEMs.

Reviewer 1 ·

Basic reporting

Overall, the manuscript is quite well written, and the authors did an excellent job of thoroughly analyzing the data and presenting the work.

One thing that could be added in the introduction is to point out the differences between two related but different experimental techniques: binocular rivalry and CFS, since findings from both are discussed. This might help explain the seemingly discrepant findings discussed in the introduction and discussion.

Figure 3 is very nice--it shows lots of data and it’s great that individual participants’ responses are shown. However, it takes some work for the reader to get oriented to it. Not sure what could be done to improve that. Maybe highlighting the difference between the two axes (low vs high reward/punishment)? One minor suggestion is having the same quantities (eg 500, 1000) marked on the x and y axes.

Raw data
I’m not sure how the final raw data will be made available, but Including a readme.txt file would be helpful to orient people downloading the data. For instance, QueryTrial values are described in the reviewer portal, so make sure that information is available for readers.

Including the analysis code would be helpful if possible (code from R, SPSS, etc). This would aid interested readers in reproducing the same computational results described in the paper. See https://kbroman.org/steps2rr/ for more info.

Including the trial number variable in the dataset would be helpful so people could reanalyze the first half vs second half analysis described in the manuscript.

Experimental design

The authors did an excellent job with very thorough analysis of their data, and look at it in multiple ways. Well done!

One omission is how they determined the sample size. A sample size of n=24 is fairly small for current practices in many areas of psychology (Button et al., 2013; though many vision studies have few participants). Especially in the context of null results, this should probably be discussed. One example of this is that one outlier seemed to dramatically change one of the findings they report. Ideally a power calculation would be performed ahead of time to determine sample size.

Validity of the findings

The analysis seems careful and thorough. Well done!

Additional comments

Overall this is a well done study that should published once some minor issues are dealt with.

One thought question is how powerful were the monetary pairings? Did participants really believe they would be compensated differently depending on the face shown? Were they compensated that way?

It sounds like only 4 facial identities were used in the study--one for each value. These faces were then shown, many time: 96 in the pre-conditioning phase, 64 times in the learning condition, and 96 times in the final post-conditioning phase. Is it possible that the 64 repetitions of each face dulled or habituated the affective impact of the faces?

Reviewer 2 ·

Basic reporting

There are several sections that are a little ambiguous and could be clearer:
- Line 149, "They were further informed that they would receive or lose the amount of money depicted in each trial" – this sentence is ambiguous.
- Line 329, Could you expand a little on the paper that does show an effect of instrumental conditioning using monetary values? This paper seems particularly relevant, and so having further discussion of it would be helpful.

Experimental design

Some aspects of the design could be more fully justified.
- Line 145, why 75% reinforcement rate?

Validity of the findings

Results:
- Why were median values selected, rather than means? Given that RTs don't tend to be normally distributed this makes sense, but the analysis is on the change scores. It would be helpful to have some justification, just as previous studies have tended to opt for mean values.

- Does accuracy in conditioning sessions predict RTs in CFS? I just wondered if the people who were least likely to respond correctly to the conditioning query trials were clustered in their RT data. I don't think this can explain the findings, but it might be helpful to show that.

- I understand why the change in response times are calculated – these are the critical measure. However, given that change scores can sometimes be deceptive, I would really value having (in supplementary materials?) the original scores for each condition. This will have no impact on the results/conclusions, it's just for transparency.

Discussion:
- Line 344, an additional possibility would be to check that the conditioning paradigm elicits a behavioural response in a conscious (control) paradigm. Have the authors considered this?

Additional comments

This paper is well-written and interesting. The response-distribution and Bayesian analyses work very well in addressing the research questions. I do have a bit of a question over the suitability of the conditioning procedure (and the lack of an additional condition showing behavioural effects using this procedure), but I think the study is an interesting addition to the literature.

Reviewer 3 ·

Basic reporting

The Authors of the study report behavioral data of a breaking continuous flash suppression (bCFS) task run on face stimuli that were paired with high- and low-monetary reward/punishment in a conditioning phase placed between a pre- and a post-conditioning phase. The hypothesis being tested is that learning the association between face identity and high/low reward/punishment may bias pre-conscious processing of face images changing their power to break into consciousness.
The results show that: 1) higher explicit ratings of participants’ motivation to learn (both positive and negative) high vs low monetary reward/punishment identities; 2) efficient learning of identity monetary value; 3) no effect of monetary value on bCFS RTs; 4) no effect of monetary value in the first or last part of the bCFS; 5) no effect of monetary value for any decile of condition-specific RTs.
The Authors interpret these finding suggesting that the classical conditioning procedure may not be strong enough to facilitate pre-conscious face processing needed to break into consciousness, or that the effect may be linked to identity recognition which happens after the stimulus has been detected.
The Authors properly describe the literature and clarify their hypotheses in the introduction and accurately discuss the results by also providing good interpretations of the lack of effects. Language is good, figures are clear (except Fig 3 see below) and the raw data are provided.
My concerns regard the way data are analyzed, and the fact that no personality measures were recorded making it hard to say whether the lack of effects is there because participants may vary on relevant personality trait that may interact with reward/punishment sensitivity.
I am confident that the Authors may solve these issues in one round of revisions.

Introduction, line 83: Please expand on the literature showing that monetary associations have an impact on conscious perception, and that they have a behavioral relevance, or state clearly that this is not known.

Line 133: correct “Participants’ task was indicate” in “Participants’ task was to indicate”.

Please cite relevant experiments that have shown that the procedure described at line 161 produces a relevant behavioral effect on pre-conscious processing. Since there is no effect for pre-conscious identity recognition, the lack of effect may be attributed to the fact that the procedure is not effective on pre-conscious processing in general, not for face identities in particular.

Line 133: correct “Participants’ task was indicate” in “Participants’ task was to indicate”.

Experimental design

Why only 4 identities? I guess this decision was based on the need to repeat the identities a number of times to reach a good learning, but what if the association is to easy and eliminates the effect of positive/negative and high and low monetary value?

Line 162 please state that a wrong response for faces associated to positive monetary values would imply no benefit.

Line 194-197: it is not clear why the Authors are correcting for 18 comparisons. These are run on different data (2 conditions per decile since they separate positive and negative conditionings). Is the problem that the deciles are correlated? Please explain.

Please clearly state that the analyses described at line 198 were run only on the index described first, and not the deciles. Or do differently otherwise.

Validity of the findings

I am not sure what is the benefit of the removal of the outlier described at line 237 and whether this post-hoc procedure has any theoretical background. I would either avoid running and reporting this or choose a criterion to eliminate outlier individuals before running the analyses.

What is the proportion of missed (i.e. not answered) trials during the bCFS blocks? Is there any difference between positive and negative identities in response accuracy during the conditioned bCFS session? I am asking this to try and understand whether, although the query trials show that identities were recognized above chance during the conditioning phase, there might be a different level of learning for positive and negative identities which may have impacted the RTs.

I seem to understand that negative identities were potentially associated to punishment wile positive ones were associated to a potential benefit for the participant. This makes both positive and negative items relevant for participants in the query blocks, but potentially different for people who want to avoid punishment and those who aim at positive rewards (put aside that positive rewards seem to increase the motivation of their participants as expected from background knowledge in the literature). Did the Authors measure individual’s sensitivity to monetary reward/punishment or other psychological trait/state that may have impacted their sensitivity to these dimensions?
If not, as it seems, what about using for this purpose individuals’ response criterion during the conditioning phase?

Do, by any chance, positive and negative identities or high-positive, low-positive, high-negative-low-negative (per each participant) differ in terms of RTs of bCFS before the conditioning? This is not shown since the Authors report the Post-Pre difference to isolate the effect of the conditioning, but baseline differences may have masked its effect.

Is there any RTs difference between positive and negative conditioned identities when the post-pre index is computed on high and low averaged together? And between high-positive vs high-negative and low-positive and low-negative ones? These are not tested. Why are positive and negative items analyzed separately both in the t-tests and decile analyses?

Figure 3 is poorly described, and I am not able to fully understand it. I ask the Authors to provide a better description of it in the text and legend.

Additional comments

See above.

---

## Round 0.2 · Minor Revisions

Dear Dr. Rothkirch,

Even though Reviewers #1 and #2 are satisfied with your changes, Reviewer #3 has some remaining minor concerns that should be carefully addressed.

In particular, I concur with Reviewer #3 doubts about the computation of the response criterion, and I agree with them in their suggestion on how to pursue an alternative approach for computing it.

On top of that, please address Reviewer #3 request for clarification on other aspects of the data analysis and of the experimental procedure.

Reviewer 1 ·

Basic reporting

The authors thoughtfully addressed the reviewer comments.

Experimental design

The authors thoughtfully addressed the reviewer comments.

Validity of the findings

The authors thoughtfully addressed the reviewer comments.

Reviewer 2 ·

Basic reporting

No comment

Experimental design

No comment.

Validity of the findings

No comment.

Additional comments

The authors have diligently addressed all of my previous concerns. I think the manuscript is suitable for publication.

Reviewer 3 ·

Basic reporting

No comment.

Experimental design

The way the response criterion was computed is problematic. Could they provide a reference for the usage of the criterion in your design? The Authors are using both reward and loss accuracies together while the formula for the criterion asks for hits and false alarms. I think they need to extract a bias for positive and a bias for negative items by using the formula on reward and loss conditions separately and then compare the two values. In reward trials hits are defined as the ability to correctly recognize reward-faces, while false alarms are trials in which loss-faces are wrongly recognized as reward ones. In loss trials hits are defined as the ability to correctly recognize loss-faces, while false alarms are trials in which reward-faces are wrongly recognized as loss ones.

Accuracy analysis is showing individual’s aggregated values for positive, negative, high and low conditions together. This limits the possibility to evaluate whether some of these conditions had a different accuracy than the other.

For the second, and last, exploratory analysis the Authors computed accuracy in different conditions but it is unclear how this was used to calculate the RTs. Are RTs weighted for the individuals’ accuracies?

Validity of the findings

Two issues are not clear to me, one concerns the actual instruction that was given to participants regarding their final payoff, the second one concerns the effects of the conditioning procedure. I am reporting these two comments to try and improve the clarity of the methods (1) and to ask the Authors to comment on a possible reason for not finding any effect of positive and negative conditioning over aware detection of faces (2).
1) Instructions to participants concerning their payoff.
The Authors first state that “Participants were instructed to passively view the stimuli and to memorize the association between the faces and the monetary values as well as possible. They were further informed that the monetary value depicted in a trial would be counted towards their overall payoff.”. They then say that in this phase (i.e. the conditioning phase) the gain/loss amount to zero, i.e. “Since unbeknown to participants each monetary value was presented equally often, the outcome for this part of the conditioning phase amounted to zero.” and conclude by saying that “Thus, participants’ outcome was solely defined by their accuracy in the query trials (see below).”.
Some lines later the Authors say that “Participants were informed that monetary reimbursement for their participation in the experiment would depend on the accuracy of their choices during these query trials. After the experiment, participants indeed received the amount of money that they accumulated during the conditioning phase”.
So, did participants thought that their payoff was “passively” obtained by looking at the pairings in the conditioning phase or due to their ability to recognize the pairings in the query trials? While the first case would be based on classical conditioning, the second is based on participants’ performance. I believe that query trials were necessary to test whether the associations were learnt, but linking the payoff to participants’ accuracy in these trials might have switched the associations from positive and negative payoffs (for reward- and loss-faces, respectively) to positive/positive ones (for recognized reward- and recognized loss-faces, respectively), see the second point. Thus, since participants’ accuracy was very high from beginning, all the faces had a “positive” value and no difference is found.

2) Effects of the conditioning procedure.
When describing the actual rule for gaining/losing money in query trials the Authors say “More specifically, a correct assignment of a monetary reward to the respective face would yield an addition of +2 € or +0.1 €, respectively, to their payoff. A wrong response to faces associated with reward did not result in a monetary gain.”. This means that “reward-faces” did not harm participants’ payoff.
“For faces associated with a monetary punishment, participants had to assign the correct monetary value to avoid a monetary loss. This means that a correct assignment of -2 € or -0.1 € to the respective face did not yield a monetary gain or loss. However, if a participant assigned a wrong value to a punishment-related face, this resulted in a loss of -2 € or -0.1 €, respectively.”.
Thus, in query trials participants could either gain money (if correctly associating identities to gain/loss) or avoid losing money (if correctly associating identity to losses). Those are two positive outcomes for both reward- and loss-faces and may have cancelled any effect of the conditioning, se my previous point.

---

## Round 0.3 · Minor Revisions

I have now received the feedback from reviewer #3, who believes that the manuscript is now suitable for publication, pending a remaining minor concern that needs to be addressed.

Please address this last concern and submit it at your earliest convenience.

Reviewer 3 ·

Basic reporting

No comment.

Experimental design

No comment.

Validity of the findings

No comment.

Additional comments

I thank the Authors for answering to my points.
In relation to their response:

"The reviewer asks what participants believed their payoff was dependent on. In the current version of the manuscript, we clarify that participants were made to believe that their payoff would depend on both, the monetary values that they passively viewed and their accuracy in the query trials (ll. 193ff): “Thus, participants expected that their payoff was dependent on both, the passively viewed pairings of monetary values and faces as well as their performance during the query trials. Note that in case of negative values participants had to respond accurately to avoid monetary losses during the query trials.”"

I guess this was done 1) to make high-valued faces more relevant in the passive viewing phase (as they implied larger gain/losses for the participant), and 2) to have participants paying attention to all stimuli in query trials (as higher accuracies imply more money) to understand whether the associations were properly learnt.
I think the Authors need to explain the rationale of their (two different?) instructions.

I believe that the manuscript should now be accepted provided the Authors include a sentence to explain this point.

---

## Round 0.4 · accepted · Accept

I am glad to inform you that the manuscript is now suited for publication on PeerJ.